# Synthesizing Verified Mathematical Problems

**Xuefeng Li**[1,2]    **Yanheng He**[1,2]    **Pengfei Liu**[1,2,3*]

[1]Shanghai Jiao Tong University
[2]Generative AI Research Lab    [3]Shanghai AI Laboratory

## Abstract

Mathematical data synthesis offers a potentially effective solution for enhancing the mathematical capabilities of large language models. However, existing methods either synthesize a large number of rationales based on existing questions, limiting the diversity of the questions, or rely on advanced proprietary models to directly generate new questions without verification, which cannot guarantee the correctness of the synthesized problems. This paper introduces a novel method, mathematical data synthesis through Algorithmic **A**bstraction, **I**mplementation, and **C**ontextualization (AIC), to synthesize new and verifiable mathematical problems. **AIC** abstracts mathematical problems into algorithms, implements these algorithms as code functions, and contextualizes them under different conditions to create new problems, which are then verified using code functions. Experimental results on multiple challenging mathematical benchmarks show that models fine-tuned on our synthesized data are superior to previous state-of-the-art models. Further experiments indicate that, when controlling for the same synthesizer, data synthesized using the AIC method is not only more accurate but also more effective at improving the model's mathematical abilities.

## 1   Introduction

Large language models (LLMs) have made significant strides, expanding from natural language processing to areas like code generation and creative writing [3, 29, 4]. Their success stems from vast amounts of high-quality training data [30, 9]. As the availability of untapped high-quality data diminishes, LLM research faces a problem of data scarcity [25]. Consequently, data synthesis, using generative models to create data similar to real data, offers a solution to this scarcity by supplementing real-world data [18, 2]. For synthetic data to be effective, it must maintain quality comparable to real data [27, 24], particularly for mathematical data, which demands high logical consistency.

Research on enhancing LLMs' mathematical abilities through instruction tuning mainly follows two approaches. The first generates rationales for known mathematical problems using LLMs, filtering rationales based on the correctness of the final answer [30, 26, 23, 22, 10], though this limits the diversity of problems. The second approach uses advanced LLMs, like GPT-4 [1], to generate new questions and rationales [6, 19, 17, 21, 13, 16], enhancing data diversity but risking accuracy without verification [28, 20]. Therefore, a method that generates new problems while ensuring their correctness is essential for producing diverse and accurate synthetic mathematical data.

In this paper, we propose Mathematical Data Synthesis via Algorithmic Abstraction, Implementation, and Contextualization (AIC). The central idea is that many mathematical problems can be addressed by abstract algorithms. By abstracting such algorithms from mathematical problems and contextualizing them, we can generate new mathematical questions and corresponding rationales. Moreover, abstract mathematical algorithms can be implemented using Python to verify the correctness of the synthesized

---

*Corresponding author.

38th Conference on Neural Information Processing Systems (NeurIPS 2024). Workshop on MATH-AI

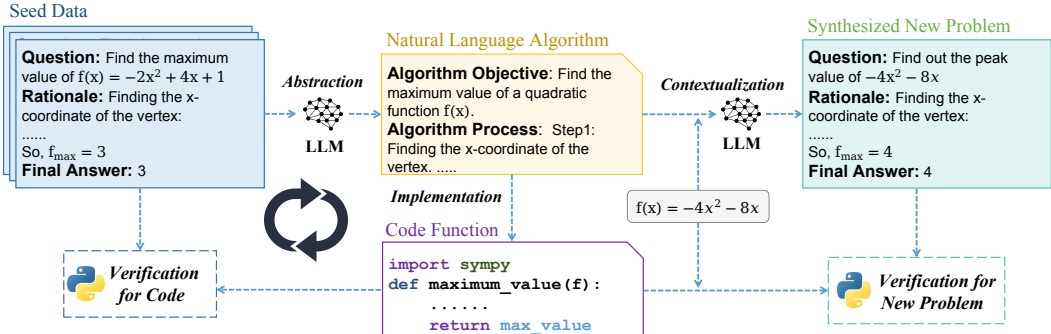

Figure 1: An Overview of AIC: (1) The synthesizer(LLM) abstracts mathematical problems (Seed Data) into natural language algorithms. (2) These algorithms are implemented in python by the synthesizer, with their correctness verified through a verification process. (3) Finally, the synthesizer contextualizes the abstract algorithms to generate new problems, employing a verification mechanism to ensure the correctness of the newly synthesized problem.

data. As shown in Figure 1, the process of AIC is divided into two stages. **Stage1 Algorithm Abstraction and Implementation**: First, we use large language models (LLMs) as a synthesizer to abstract existing mathematical problems, which serve as seed data. Each entry includes the question, rationale, and final solution, and is transformed into a natural language algorithm. Next, we prompt the synthesizer to implement the algorithm as a Python code function and verify its correctness using a verification mechanism. **Stage2 Algorithm Contextualization**: The synthesizer contextualizes the natural language algorithm and generates a new mathematical problem. Then, the conditions of this newly generated problem are fed into the corresponding code function, and by checking if the final answer generated by the synthesizer aligns with the result of the code execution, thereby verifying the correctness of the synthesized problem.

We evaluated the model on several challenging mathematics benchmarks, including MATH [12], MathOdyssey [8], finding that data synthesized using AIC can significantly improve the performance of the synthesis model itself and is highly competitive compared to other methods. AIC not only has the capability to synthesize a large volume of high-quality mathematical data but also paves a new way for generating verifiable mathematical problems.

## 2 Methods

In this paper, we propose a data synthesis method for generating new mathematical problems with verified solutions. Our method comprises two stages. In the first stage, we employ an LLM as a synthesizer to abstract algorithms from existing mathematical problems. We then prompt the synthesizer to implement these algorithms as Python code functions and verify the code's correctness. In the second stage, the synthesizer contextualizes the algorithms into new mathematical problems, using the code functions to verify the correctness of the synthesized problems.

Let $\mathcal{D}_{\text{seed}} = (q_i, r_i, a_i)_{i=1}^N$ represent a typical mathematical training dataset, which serves as seed data, where $q_i, r_i, a_i$ are question, rationale and final answer of the i-th problem. In addition to the seed data, the synthesis process utilizes large language models such as $\mathcal{M}$ (e.g., Mixtral-8×7B-Instruct [15], Llama3-70B-Instruct [7]) and a code interpreter $\mathcal{C}$.

### 2.1 Stage1: Algorithm Abstraction and Implementation

For each piece of data $d_i = (q_i, r_i, a_i)$, the LLM is asked to first analyze the question $q_i$ and the rationale $r_i$ to understand the core goals of the problem, identify the key operations and steps of reasoning, determine the sequential relationships between steps, and finally, identify the mathematical objects such as integers, series and expressions in question $q_i$ that are independent of the core steps in reasoning, parameterizing them as placeholder variables. This way, a question $q_i$ and its rationale $r_i$ can be transformed into an **algorithm objective** $o_i$ and an **algorithm process** $p_i$. As show in Figure 2, We also prompt the synthesizer $\mathcal{M}$ to generate additional information about the algorithm,

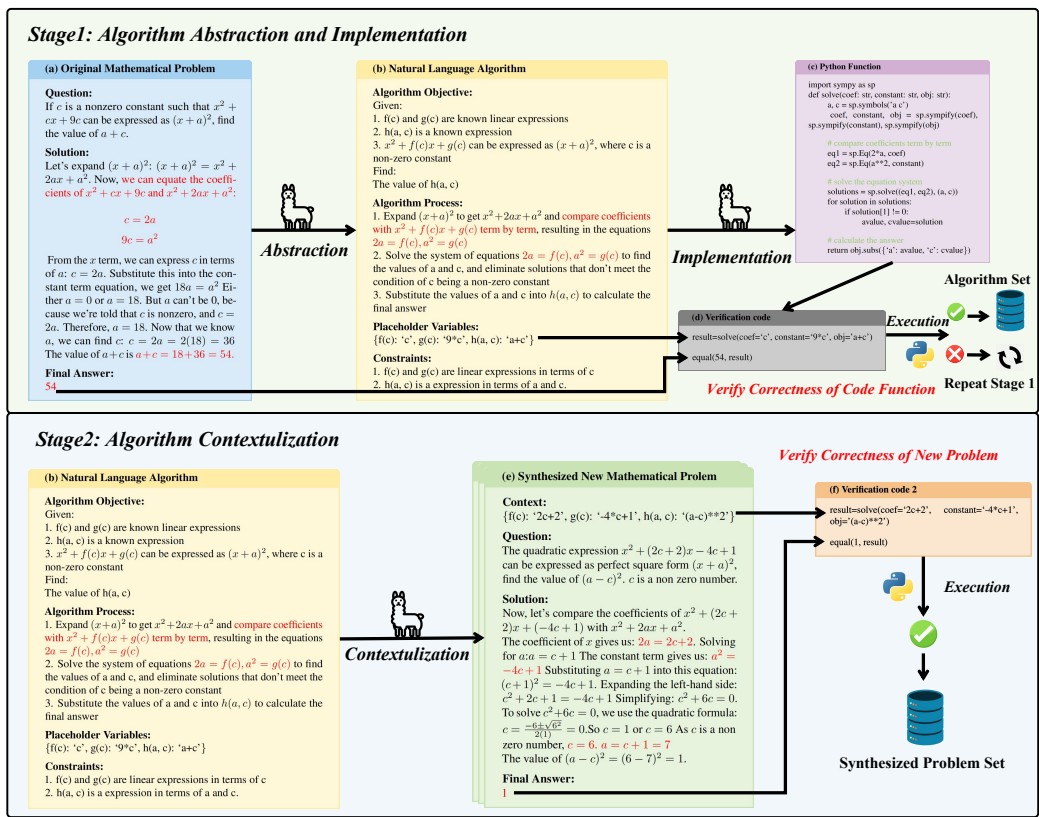

Figure 2: A more detailed overview of the synthesis pipeline.

including **placeholder constraint** $c_i$, which specify the placeholder variables' types, value ranges and relationships with other placeholder variables; **placeholder values** $v_i$, which indicate the values of the placeholder variables in the original problem. Overall, we prompt the synthesizer $\mathcal{M}$ to abstract a mathematical problem $d_i = (q_i, r_i, a_i)$ into a natural language algorithm $\Psi_i$.

For each algorithm $\Psi_i$, the synthesizer $\mathcal{M}$ programs a code function $f_i$ by Python, where the parameters are the placeholder variables, and the return value is the final result of the algorithm. Furthermore, to ensure the correctness of the algorithm and function, we propose a verification mechanism, using the original problem $d_i = (q_i, r_i, a_i)$ as a test case, inputting placeholder values $v_i$ into the function $f_i$, obtaining the function's return value and comparing it with the original answer $a_i$, to filter out incorrect functions. If verification fails, we regenerate the algorithm for the problem, repeating until the algorithm passes the verification or reaches the maximum number of iterations $I$.

## 2.2 Stage2: Algorithm Contextualization

Contextualization aims to transform abstract natural language algorithms into specific mathematical problems. For any given algorithm $\Psi_i = (o_i, p_i, v_i, c_i)$ and the corresponding code function $f_i$, first, the synthesizer generates $K$ possible values of **placeholder variable** based on the algorithm, denote as $pv_i^j, j = 1, ..., K$, which assigning specific mathematical objects that comply with the algorithm's constraints, which can also be referred to as a **context**. With the algorithm and placeholder variables in place, the synthesizer generates specific mathematical question $q_i^j$, corresponding rationale $r_i^j$, and final answer $a_i^j$ for the algorithm $\Psi_i$ in the current context $pv^j$.

Unlike traditional synthesis algorithms that lack verification, here we can input the placeholder variable values $pv_i^j$ into the code function $f_i$ and execute it by code interpreter $\mathcal{C}$, filtering out incorrect synthesized data by checking whether the execution result $g_i^j$ matches the final answer $a_i^j$ given by the synthesizer through algorithm contextualization. By generating numerous contexts, an algorithm can be contextualized into many mathematical instruction-tuning data points.

# 3 Experimental Results

## 3.1 Experiments Setting

**Data**   We use the training set from MATH [12] and a small subset of MAmmoTH2 [31] including about 30,000 data points as seed data. subset includes approximately 30,000 data points. Data synthesis is conducted using Mixtral-8×7B-Instruct and Llama3-70B-Instruct, resulting in the AIC-M and AIC-L datasets, respectively. Further details about the data are provided in Appendix A.

**Training**   We follow a standard supervised fine-tuning approach to train several models, including Mistral-7B-Base [14], Llama3-8B-Base, Mixtral-8×7B-Instruct [15], and Llama3-70B-Instruct.

**Evaluation**   We evaluate the effectiveness of our method using five high-difficulty mathematical benchmarks, including the in-domain benchmark MATH and out-of-domain benchmarks GaoKaoBench-Math [32], MathOdyssey [8], OlympiadBench-Math [11], and TheoremQA [5].

More detailed information on the experimental settings is provided in Appendix B.

## 3.2 Effetiveness of Synthesized Data

Table 1: Mixtral-AIC and Llama3-70B-AIC refer to models trained using AIC-M on Mixtral-8×7B-Instruct and AIC-L on Llama3-70B-Instruct, respectively. On the other hand, Mixtral-Seed and Llama3-70B-Seed are models trained with the seed data on corresponding models.

| Model | MATH | GaoKao | Odyssey | Olypaid | TheoremQA | Avg |
|---|---|---|---|---|---|---|
| | | | Mixtral-8×7B-Instruct | | | |
| Mixtral-Seed | 27.6 | 16.9 | 8.7 | 6.8 | 12.8 | 14.6 |
| Mixtral-AIC | $35.4_{+7.8}$ | $19.0_{+2.1}$ | $12.6_{+3.9}$ | $9.9_{+3.1}$ | $13.8_{+1.0}$ | $18.1_{+3.5}$ |
| | | | Llama3-70B-Instruct | | | |
| Llama3-70B-Seed | 39.2 | 22.0 | 9.5 | 10.5 | 15.0 | 19.2 |
| Llama3-70B-AIC | $48.7_{+9.5}$ | $30.5_{+8.5}$ | $14.4_{+4.9}$ | $15.1_{+4.6}$ | $18.3_{+3.3}$ | $25.4_{+6.2}$ |

We separately fine-tune Mixtral-8×7B-Instruct and Llama3-70B-Instruct using data synthesized by Mixtral-87B-Instruct (AIC-M) and Llama3-70B-Instruct (AIC-L) to evaluate whether the synthesized data can enhance the performance of the models. Since the data synthesis process involves both large language models (LLMs) and seed data, we compared the performance of models trained with both synthesized and seed data. As shown in Table 1, training with the synthesized data significantly outperforms training with the original seed data, demonstrating the effectiveness of our approach.

## 3.3 Comparison with other model

Table 2: Comparison of different models testing accuracy on mathematical benchmarks.

| Model | Synthesis Model | MATH | GaoKao | Odyssey | Olypaid | TheoremQA | Avg |
|---|---|---|---|---|---|---|---|
| Mistral-7B-WizardMATH | GPT4 | 32.3 | - | - | - | - | - |
| Mistral-7B-MetaMATH | GPT3.5 | 27.7 | 14.9 | 5.9 | 6.5 | 6.0 | 13.1 |
| Mistral-7B-MMIQC | GPT4 | 31.5 | 17.9 | 7.2 | 6.8 | 9.2 | 14.5 |
| Mistral-7B-MathScale | GPT4 | 35.2 | - | - | - | - | - |
| Mistral-7B-AIC | Llama3 | $36.4_{+1.2}$ | $20.8_{+2.9}$ | $8.7_{+1.5}$ | $11.1_{+4.5}$ | $12.5_{+2.7}$ | $17.9_{+3.4}$ |
| Llama3-8B-MetaMATH | GPT4 | 31.5 | 14.7 | 6.4 | 6.8 | 10.2 | 13.9 |
| Llama3-8B-MAmmoTH2 | GPT4 | 35.8 | - | - | - | - | |
| Llama3-8B-MMIQC | GPT4 | 37.5 | 15.3 | 11.3 | 6.9 | 9.7 | 16.1 |
| Llama3-8B-AIC | Llama3 | $39.0_{+1.5}$ | $20.6_{+5.3}$ | $10.8_{-0.5}$ | $8.8_{+1.9}$ | $11.6_{+1.4}$ | $18.1_{+2.0}$ |

In this section, we train two base models Mistral-7B-Base and Llama3-8B-Base using AIC-L and compare them with other models, including WizardMath, MetaMath, MMIQC, MathScale, and MAmmoTH2. Additional details about baselines are provided in Appendix B.3.

Table 2 presents the performance of our method compared to other data synthesis approaches across various high-difficulty math benchmarks. Among models based on Mistral-7B-Base, Mistral-7B-

AIC demonstrated an average improvement of 3.4%. For models derived from Llama3-8B-Base, Llama3-8B-AIC showed an improvement of 2.0%. Additionally, while most competing models utilize closed-source advanced models (e.g., GPT-3.5, GPT-4) for data synthesis, our approach leverages the open-source model, further underscoring the effectiveness of our method.

## 3.4 Fair Comparison with Other Methods

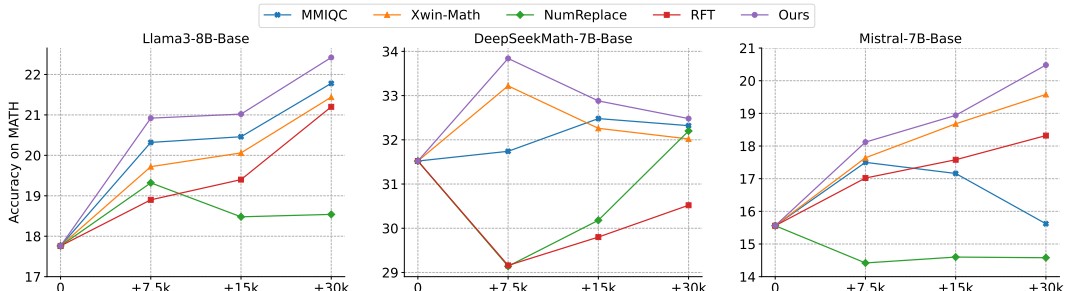

Figure 3: Fair comparison with other methods.

Given the variation in both the models used for data synthesis and the scale of data synthesis across the comparison objects in Section 3.3, these differences do not accurately represent the strengths and weaknesses of the synthesis methods. To address this, we standardize both the data synthesis models and the scale of data synthesis in this section, allowing for a more comprehensive evaluation of the methods. We chose MATH as the seed data and conducted all evaluations on MATH. The methods compared include NumReplace, MMIQC, and Xwin-MATH, as introduced in Appendix B.3. We trained models on various data scales to thoroughly assess the effectiveness of the synthesis methods.

The results in Figure 3 show that our method outperforms the baselines at any scale. We believe this is because, when generating more difficult problems, methods without a verification mechanism often lead to errors in the synthesized data, thereby reducing its quality.

## 3.5 Effectiveness of Verification

We investigate the verification mechanism's effectiveness by comparing two equal-sized datasets: one before and one after its application. For simplicity, this experiment uses only MATH as the seed data and test set.

The results in Table 3, demonstrate that the verification mechanism enhances model performance. This improvement stems from the fact that the final answers generated by the code are generally correct, and filtering the rationale based on these answers improves the logical and computational accuracy of the data, thereby enhancing its overall quality. These findings highlight the importance of verifying the correctness of synthesized data.

Table 3: Ablation Study on the verification mechanism.

| Verification | Samples | MATH |
|---|---|---|
| Llama3-8B-Base | | |
| ✓ | 34k | 21.2$_{+1.1}$ |
| ✗ | 34k | 20.1 |
| Mistral-7B-Base | | |
| ✓ | 34k | 18.8$_{+1.0}$ |
| ✗ | 34k | 17.8 |

## 4 Conclusion

The paper proposes a mathematical synthesis approach generating diverse, verified synthetic data through algorithmic abstraction and contextualization, offering a scalable solution for enhancing LLM mathematical capability.

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

# Appendix

## A  Data

### A.1  Seed Data

We use the MATH training set and a small subset of MAmmoTH2 as the seed data for data synthesis. The MATH dataset consists of competition-level math problems, covering a wide range of topics such as algebra, geometry, probability, number theory, and more. MAmmoTH2, on the other hand, is an instruction-tuned dataset created by retrieving, cleaning, and rewriting mathematical content from the internet, containing a large number of math problems.

The MATH training set contains 7,500 problems, and we selected all of them as seed data. MAmmoTH2 consists of 10 million entries, of which 2 million have been open-sourced. From this, we selected 30,000 high-quality examples to be used as seed data. We applied LLMs to filter the data, prioritizing high quality and the presence of correct answers, and included these filtered examples in the seed data.

### A.2  Synthesized Data

We used two models, Llama3-70B-Instruct and Mixtral-8×7B-Instruct, to synthesize the data. The total amount of data, as well as the data from each type of seed data, is shown in Table 4.

Table 4: The statistics of synthesized data.

| Synthesis Model | Total | MATH | MAmmoTH2-Subset |
|---|---|---|---|
| Llama3-70B-Instruct | 1670k | 970k | 700k |
| Mixtral-8×7B-Instruct | 1350k | 700k | 650k |

## B  Experiments Setting

### B.1  Training

When performing standard supervised fine-tuning, we used the template shown in Figure 4, whether fine-tuning the Base model or the Instruct model. This is because we found that using the native Instruct template or the new template made almost no difference in model performance after training.

Figure 4: Template for supervised fine-tuning.

**Training Template**

Question:{Question}
Answer: Let's think Step by Step.
{Rationale}
#### {Final Answer}

For all models, we applied full-parameter fine-tuning using the Adam optimizer, with a warmup ratio set to 0.1 and the learning rate scheduler set to a cosine scheduler. The values for the number of epochs, learning rate, and batch size vary depending on the model and are shown in Table 5.

### B.2  Benchmarks

We selected five high-difficulty benchmarks, which include the MATH test set, GaoKao-MATH, OlympiadBench-MATH, MathOdyssey, and TheoremQA.

Table 5: The statistics of synthesized data.

| Model | Batch size | Epoch | Learning Rate |
|---|---|---|---|
| Mistral-7B-Base | 128 | 3 | 2e-6 |
| Llama3-8B-Base | 128 | 1 | 1e-5 |
| Mixtral-8×7B-Instruct | 128 | 3 | 1e-5 |
| Llama3-70B-Instruct | 128 | 1 | 1e-5 |

- The MATH test set is distributed similarly to the MATH training set, featuring high difficulty and wide coverage.
- GaoKao-MATH contains 5000 pieces of math problems from China's Gaokao (college entrance examination).
- MathOdyssey consists of 387 pieces of professional math problems from both university and high school levels, serving as the problem set for the 2024 Global AI Competition (GAIC) math contest.
- OlympiadBench-MATH consists of 675 pieces of Olympiad-level math competition problems. We selected only the pure text-based math problems from OlympiadBench.
- TheoremQA includes 800 problems from various fields such as mathematics, physics, and economics, which require domain-specific theorems to solve.

### B.3 Baselines

Our comparisons focus on various methods for synthesizing mathematical instruction-tuning data and the corresponding models, including WizardMATH, MetaMATH, MMIQC, MAmmoTH2, MATHScale, and Xwin-MATH.

- WizardMATH enhances existing data using the Eval-Instruct method, which includes increasing and decreasing the difficulty. Additionally, WizardMATH employs reinforcement learning to further improve model performance.
- MetaMATH introduces methods for rephrasing questions and backward reasoning to expand the existing data.
- MMIQC iterates on existing questions to increase their complexity, generating new and more challenging questions.
- MAmmoTH2 retrieves, cleans, and rewrites mathematical content from the internet to create instructional question data.
- MATHScale extracts knowledge points from existing data and uses these points as seed information for large-scale data synthesis.
- Xwin-MATH directly requires the LLM to generate a completely new math problem based on an existing one.

Since MetaMATH and MMIQC have open-sourced their data, the experimental results for MetaMATH and MMIQC were obtained from our own training and testing. For MMIQC, we used only the synthetic data for training and did not use StackExchange data. For WizardMATH, MAmmoTH2, MATHScale, and Xwin-MATH, since the data was not open-sourced, we compared our results directly with those reported in the corresponding papers.

### B.4 Evaluation

In evaluation, we use greedy decoding and apply the same template as used in training. We evaluate using exact match criteria, where only results that are either exactly the same as the correct answer or meet certain rules for matching the correct answer are considered correct.

## C   Synthesized Examples

We provide an example of synthesized new problem, along with its original mathmatical problem, natural language algorithm and code function.

**Original Mathematical Problem**

**Question:**
A printer prints 17 pages per minute. How many minutes will it take to print 200 pages? Express your answer to the nearest whole number.

**Solution:**
$\frac{200 \text{ pages}}{17 \text{ pages per minute}} \approx \boxed{12}$ minutes.

**Final Answer:**
12

---

**Natural Language Algorithm**

**Algorithm Objective**
Given:
- A printer prints $p$ pages per minute.
- The desired number of pages to be printed is $n$.
Find:
- Approximate number of minutes needed to print $\boxed{n}$ pages.

**Algorithm Process**
1. **Calculate the Number of Minutes Needed:**

$$\text{Number of minutes} = \frac{\text{Total number of pages}}{\text{Pages per minute}}$$

$$\text{Number of minutes} = \frac{n}{p}$$

**Placeholder Variables:**
{"p": 17, "n": 200}

**Constraints:**
None 12

---

**Code Function**

```
def print_time(p, n):
    minutes = round(n / p)
    return minutes
```

---

**Synthesized New Problem**

**Question:**
A printer produces 24 pages per minute. How many minutes will it take to print 240 pages? Round your answer to the nearest whole number.

**Solution:**
240/24=10. So it will take 10 minute to print 240 pages.

**Final Answer:**
10

## D   Compute Resources

For algorithm abstraction and implementation, the required GPU time depends on the maximum number of iterations. When the iteration count is set to 50, it requires about 20 hours of runtime on 8*A100 machines. For contextualization, it depends on the number of generated problems. To generate 100,000 problems, it takes approximately 20 hours on 8*A100 machines.

# E   Limitations and Future Work

Our method is limited to mathematical data synthesis and needs to be further extended to other types of inference data. Additionally, our method lacks diversity in problem generation, which requires defining more meta-level algorithms and proposing corresponding algorithm abstraction methods and validation mechanisms to improve the diversity of both algorithms and generated problems.

