# OpenReview forum: "Synthesizing Verified Mathematical Problems"
_NeurIPS.cc/2024/Workshop/MATH-AI — MATH-AI 24_

### Official Review · Reviewer_ckUL · 2024-10-04
**Good paper "Synthesizing Verified Mathematical Problems"**

**Rating:** 7
**Confidence:** 4

**Review:**

This paper proposes a novel approach to enhancing the math capabilities of LLMs through a method called Mathematical Data Synthesis via Algorithmic Abstraction, Implementation, and Contextualization (AIC). This approach generates diverse and accurate mathematical problems by abstracting mathematical problems into algorithms, implementing them as Python code functions, and contextualizing them to create new problems. The correctness of these synthesized problems is verified using code. The method addresses the limitations of current LLM-based mathematical data synthesis, including restricted data diversity and accuracy issues. The paper demonstrates that this method significantly improves the performance of models when tested on various high-difficulty mathematical benchmarks.

---
## Strength
- **High-quality data with verification:** The use of Python code to verify the correctness of synthesized mathematical problems adds a layer of reliability to the data, ensuring high accuracy.
- **Performance Improvement:** This approach shows significant performance improvements in solving mathematical benchmarks compared to other models, as highlighted in their comprehensive experimental results.
- **Potential in scalability:** The method allows for scalable synthesis of mathematical problems, offering a potential solution to the scarcity of high-quality mathematical training data.

---
## Weakness:
- **Limited Problem Diversity:** The authors acknowledge a limitation in the diversity of generated problems, particularly in defining more meta-level algorithms and improving validation mechanisms. Moreover, I am curious about whether their finetuned model generalize well to  other reasoning problems.
- **Concern on Catastrophic Forgetting:** This paper only prove that their finetuning process enhance math capabilities, while do not benchmark the general capabilities of these models. I think the general capabilities are equally important to solving math problems for a chatbot assistant. The authors may evaluate their models on MMLU or MT-Bench for a more comprehensive understanding to their models.

---
## Suggested Improvement:
- Typos: Line 136
- Evaluation on general capabilities.

---

### Official Review · Reviewer_ESQX · 2024-10-05
**This paper proposes a novel method called Mathematical Data Synthesis through Algorithmic Abstraction, Implementation, and Contextualization (AIC) for generating high-quality synthetic mathematical problems and solutions. The key idea is to abstract existing math problems into algorithmic templates, implement them as code, and then contextualize them to create new verified problems. The authors evaluate AIC on several challenging math benchmarks and show it can significantly improve the mathematical reasoning capabilities of large language models.**

**Rating:** 8
**Confidence:** 4

**Review:**

Pros:

Novel and well-motivated approach: The AIC method addresses key limitations of existing math data synthesis approaches by generating diverse new problems while ensuring correctness.

Rigorous methodology: The two-stage process of abstraction/implementation and contextualization is clearly explained and well-designed to achieve the goal of verified data synthesis.

Strong empirical results: Experiments on multiple challenging benchmarks demonstrate significant performance gains over competitive baselines and existing synthesis methods.

Reproducibility: The authors provide detailed experimental settings and open-source their code to enable reproducibility.
Broader impact: This work has potential to meaningfully advance mathematical reasoning capabilities of AI systems.

Cons:

Limited theoretical analysis: While the empirical results are strong, the paper lacks theoretical analysis of the proposed method's properties or guarantees.

Scalability concerns: The computational requirements for large-scale data synthesis using this method are quite high, which may limit its practical applicability.

Narrow scope: The method is specifically designed for mathematical problem synthesis and may not generalize well to other domains.

Benchmark selection: While the chosen benchmarks are relevant, including additional diverse math tasks could further strengthen the evaluation.

Human evaluation: Incorporating human evaluation of the synthesized problems' quality could provide valuable additional validation.

Overall Evaluation:
This paper presents a novel and effective approach to mathematical data synthesis that demonstrates strong empirical results on challenging benchmarks. The clear methodology, reproducible experiments, and potential impact on advancing mathematical reasoning in AI make this a valuable contribution to the field. While there are some limitations in terms of theoretical analysis and broader applicability, the strengths of the work outweigh these concerns.
The paper is well-written, with a clear presentation of the method and results. The authors provide sufficient detail on the experimental setup and make their code available, which is commendable for reproducibility. The extensive comparisons with existing methods and baselines strengthen the empirical validation.
One particularly strong aspect is how the method addresses the key challenge of generating diverse new problems while ensuring correctness - a limitation of many existing approaches. The use of algorithmic abstraction and code-based verification is an innovative solution to this problem.
The results showing significant improvements over state-of-the-art baselines on multiple challenging benchmarks are impressive and suggest this method could have meaningful impact on advancing mathematical reasoning capabilities in AI systems.
While the paper focuses primarily on empirical results, some additional theoretical analysis of the method's properties or guarantees could further strengthen the work. The high computational requirements may also limit practical scalability, though this is somewhat mitigated by the strong performance gains demonstrated.
Overall, this paper makes a valuable contribution to an important problem in AI and machine learning.

---

### Official Review · Reviewer_QjV9 · 2024-10-07
**Paper introduces a novel data synthesis method with built-in correctness verification,  enhancing model performance, and is recommended for acceptance with minor revisions to address the identified weaknesses and questions.**

**Rating:** 7
**Confidence:** 4

**Review:**

Summary:
This work explores high quality data generation by abstracting mathematical problems into algorithms, implements these algorithms to code for verification and contextualizing them to generate new problem statements. The method is tested on in domain and out of domain benchmarks and show improvement on these baselines. This work contributes to open source data synthesis for model improvement by providing a framework for generating verified problems at scale.

Strengths:
1. This paper shows that models trained with synthesized data using AIC outperform those with seed data alone on both out of domain and in domain datasets.
2. With a built in verification mechanism of the precision of code execution to ensure correctness of the generated data, they address a key limitation of existing approaches of generating diverse new problems while ensuring correctness.

Weakness:
1. The verification method solely focusses on correctness of the final answer which can be limited due to logical fallacies and logical consistency in the generated data points.
2. A few grammatical errors and typos can be addressed.
* Line 52 -  2.1 Stage1: Alogorithm,
* Line 71- 2.2 Stage2: Alogorithm,
* Line 30 - we cat



Questions:

1. How do you ensure that the new problems are sufficiently different from the seed data and not just minor variations?
2. Have you considered extending the verification mechanism to check the logical consistency and correctness of the reasoning steps in the rationales, not just the final answers? How might this be implemented to ensure that the generated data points are free from logical fallacies?

---

### Decision · Program_Chairs · 2024-10-08

Accept